# Truncated Variance Reduction: A Unified Approach to Bayesian Optimization and Level-Set Estimation

**Ilija Bogunovic[1], Jonathan Scarlett[1], Andreas Krause[2], Volkan Cevher[1]**
[1] Laboratory for Information and Inference Systems (LIONS), EPFL
[2] Learning and Adaptive Systems Group, ETH Zürich

{ilija.bogunovic,jonathan.scarlett,volkan.cevher}@epfl.ch, krausea@ethz.ch

## Abstract

We present a new algorithm, truncated variance reduction (TRUVAR), that treats Bayesian optimization (BO) and level-set estimation (LSE) with Gaussian processes in a unified fashion. The algorithm greedily shrinks a sum of truncated variances within a set of potential maximizers (BO) or unclassified points (LSE), which is updated based on confidence bounds. TRUVAR is effective in several important settings that are typically non-trivial to incorporate into myopic algorithms, including pointwise costs and heteroscedastic noise. We provide a general theoretical guarantee for TRUVAR covering these aspects, and use it to recover and strengthen existing results on BO and LSE. Moreover, we provide a new result for a setting where one can select from a number of noise levels having associated costs. We demonstrate the effectiveness of the algorithm on both synthetic and real-world data sets.

## 1 Introduction

Bayesian optimization (BO) [1] provides a powerful framework for automating design problems, and finds applications in robotics, environmental monitoring, and automated machine learning, just to name a few. One seeks to find the maximum of an unknown reward function that is expensive to evaluate, based on a sequence of suitably-chosen points and noisy observations. Numerous BO algorithms have been presented previously; see Section 1.1 for an overview.

Level-set estimation (LSE) [2] is closely related to BO, with the added twist that instead of seeking a maximizer, one seeks to classify the domain into points that lie above or below a certain threshold. This is of considerable interest in applications such as environmental monitoring and sensor networks, allowing one to find all "sufficiently good" points rather than the best point alone.

While BO and LSE are closely related, they are typically studied in isolation. In this paper, we provide a unified treatment of the two via a new algorithm, *Truncated Variance Reduction* (TRUVAR), which enjoys theoretical guarantees, good computational complexity, and the versatility to handle important settings such as pointwise costs, non-constant noise, and multi-task scenarios. The main result of this paper applies to the former two settings, and even the fixed-noise and unit-cost case, we refine existing bounds via a significantly improved dependence on the noise level.

### 1.1 Previous Work

Three popular myopic techniques for Bayesian optimization are expected improvement (EI), probability of improvement (PI), and Gaussian process upper confidence bound (GP-UCB) [1, 3], each of which chooses the point maximizing an acquisition function depending directly on the current posterior mean and variance. In [4], the GP-UCB-PE algorithm was presented for BO, choosing the highest-variance point within a set of potential maximizers that is updated based on confidence bounds. Another relevant BO algorithm is BaMSOO [5], which also keeps track of potential maximizers, but instead chooses points based on a global optimization technique called simultaneous

online optimization (SOO). An algorithm for level-set estimation with GPs is given in [2], which keeps track of a set of unclassified points. These algorithms are computationally efficient and have various theoretical guarantees, but it is unclear how best to incorporate aspects such as pointwise costs and heteroscedastic noise [6]. The same is true for the Straddle heuristic for LSE [7].

Entropy search (ES) [8] and its predictive version [9] choose points to reduce the uncertainty of the location of the maximum, doing so via a *one-step lookahead* of the posterior rather than only the current posterior. While this is more computationally expensive, it also permits versatility with respect to costs [6], heteroscedastic noise [10], and multi-task scenarios [6]. A recent approach called minimum regret search (MRS) [11] also performs a look-ahead, but instead chooses points to minimize the regret. To our knowledge, no theoretical guarantees have been provided for these.

The multi-armed bandit (MAB) [12] literature has developed alongside the BO literature, with the two often bearing similar concepts. The MAB literature is far too extensive to cover here, but we briefly mention some variants relevant to this paper. Extensive attention has been paid to the *best-arm identification* problem [13], and cost constraints have been incorporated in a variety of forms [14]. Moreover, the concept of "zooming in" to the optimal point has been explored [15]. In general, the assumptions and analysis techniques in the MAB and BO literature are quite different.

## 1.2 Contributions

We present a unified analysis of Bayesian optimization and level-set estimation via a new algorithm Truncated Variance Reduction (TRUVAR). The algorithm works by keeping track of a set of potential maximizers (BO) or unclassified points (LSE), selecting points that shrink the uncertainty within that set up to a truncation threshold, and updating the set using confidence bounds. Similarly to ES and MRS, the algorithm performs a one-step lookahead that is highly beneficial in terms of versatility. However, unlike these previous works, our lookahead avoids the computationally expensive task of averaging over the posterior distribution and the observations.

Also in contrast with ES and MRS, we provide theoretical bounds for TRUVAR characterizing the cost required to achieve a certain accuracy in finding a near-optimal point (BO) or in classifying each point in the domain (LSE). By applying this to the standard BO setting, we not only recover existing results [2, 4], but we also strengthen them via a significantly improved dependence on the noise level, with better asymptotics in the small noise limit. Moreover, we provide a novel result for a setting in which the algorithm can choose the noise level, each coming with an associated cost.

Finally, we compare our algorithm to previous works on several synthetic and real-world data sets, observing it to perform favorably in a variety of settings.

## 2 Problem Setup and Proposed Algorithm

**Setup:** We seek to sequentially optimize an unknown reward function $f(x)$ over a finite domain $D$.[1] At time $t$, we query a single point $x_t \in D$ and observe a noisy sample $y_t = f(x_t) + z_t$, where $z_t \sim N(0, \sigma^2(x_t))$ for some known noise function $\sigma^2(\cdot) : D \to \mathbb{R}_+$. Thus, in general, some points may be noisier than others, in which case we have *heteroscedastic noise* [10]. We associate with each point a *cost* according to some known cost function $c : D \to \mathbb{R}_+$. If both $\sigma^2(\cdot)$ and $c(\cdot)$ are set to be constant, then we recover the standard homoscedastic and unit-cost setting.

We model $f(x)$ as a Gaussian process (GP) [16] having mean zero and kernel function $k(x, x')$, normalized so that $k(x, x) = 1$ for all $x \in D$. The posterior distribution of $f$ given the points and observations up to time $t$ is again a GP, with the posterior mean and variance given by [10]

$$\mu_t(x) = \mathbf{k}_t(x)^T (\mathbf{K}_t + \mathbf{\Sigma}_t)^{-1} \mathbf{y}_t \tag{1}$$

$$\sigma_t(x)^2 = k(x, x) - \mathbf{k}_t(x)^T (\mathbf{K}_t + \mathbf{\Sigma}_t)^{-1} \mathbf{k}_t(x), \tag{2}$$

where $\mathbf{k}_t(x) = [k(x_i, x)]_{i=1}^t$, $\mathbf{K}_t = [k(x_t, x_{t'})]_{t,t'}$, and $\mathbf{\Sigma}_t = \mathrm{diag}(\sigma^2(x_1), \ldots, \sigma^2(x_t))$. We also let $\sigma_{t-1|x}^2(\overline{x})$ denote the posterior variance of $\overline{x}$ upon observing $x$ along with $x_1, \cdots, x_{t-1}$.

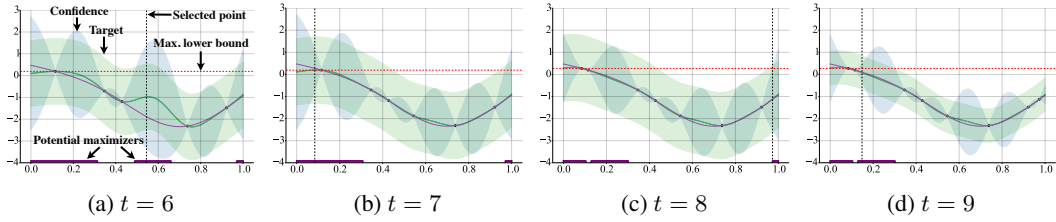

| (a) $t = 6$ | (b) $t = 7$ | (c) $t = 8$ | (d) $t = 9$ |

Figure 1: An illustration of the TRUVAR algorithm. In (a), (b), and (c), three points within the set of potential maximizers $M_t$ are selected in order to bring the confidence bounds to within the target range, and $M_t$ shrinks during this process. In (d), the target confidence width shrinks as a result of the last selected point bringing the confidence within $M_t$ to within the previous target.

We consider both *Bayesian optimization*, which consists of finding a point whose function value is as high as possible, and *level-set estimation*, which consists of classifying the domain according into points that lie above or below a given threshold $h$. The precise performance criteria for these settings are given in Definition 3.1 below. Essentially, after spending a certain cost we report a point (BO) or a classification (LSE), but there is no preference on the values of $f(x_t)$ for the points $x_t$ chosen before coming to such a decision (in contrast with other notions such as cumulative regret).

**TRUVAR algorithm:** Our algorithm is described in Algorithm 1, making use of the updates described in Algorithm 2. The algorithm keeps track of a sequence of unclassified points $M_t$, representing potential maximizers for BO or points close to $h$ for LSE. This set is updated based on the confidence bounds depending on constants $\beta_{(i)}$. The algorithm proceeds in epochs, where in the $i$-th epoch it seeks to bring the confidence $\beta_{(i)}^{1/2}\sigma_t(x)$ of points within $M_t$ below a target value $\eta_{(i)}$. It does this by greedily minimizing the sum of truncated variances $\sum_{\overline{x} \in M_{t-1}} \max\{\beta_{(i)}\sigma_{t-1|x}^2(\overline{x}), \eta_{(i)}\}$ arising from choosing the point $x$, along with a normalization and division by $c(x)$ to favor low-cost points. The truncation by $\eta_{(i)}$ in this decision rule means that once the confidence of a point is below the current target value, there is no preference in making it any lower (until the target is decreased). Once the confidence of every point in $M_t$ is less than a factor $1 + \overline{\delta}$ above the target value, the target confidence is reduced according to a multiplication by $r \in (0, 1)$. An illustration of the process is given in Figure 1, with details in the caption.

For level-set estimation, we also keep track of the sets $H_t$ and $L_t$, containing points believed to have function values above and below $h$, respectively. The constraint $x \in M_{t-1}$ in (5)–(7) ensures that $\{M_t\}$ is non-increasing with respect to inclusion, and $H_t$ and $L_t$ are non-decreasing.

---

**Algorithm 1** Truncated Variance Reduction (TRUVAR)

---

**Input:** Domain $D$, GP prior $(\mu_0, \sigma_0, k)$, confidence bound parameters $\overline{\delta} > 0$, $r \in (0, 1)$, $\{\beta_{(i)}\}_{i \geq 1}$, $\eta_{(1)} > 0$, and for LSE, level-set threshold $h$
1: Initialize the epoch number $i = 1$ and potential maximizers $M_{(0)} = D$.
2: **for** $t = 1, 2, \ldots$ **do**
3:     Choose

$$x_t = \underset{x \in D}{\arg\max} \frac{\sum_{\overline{x} \in M_{t-1}} \max\{\beta_{(i)}\sigma_{t-1}^2(\overline{x}), \eta_{(i)}^2\} - \sum_{\overline{x} \in M_{t-1}} \max\{\beta_{(i)}\sigma_{t-1|x}^2(\overline{x}), \eta_{(i)}^2\}}{c(x)}. \quad (3)$$

4:     Observe the noisy function sample $y_t$, and update according to Algorithm 2 to obtain $M_t$, $\mu_t, \sigma_t, l_t$ and $u_t$, as well as $H_t$ and $L_t$ in the case of LSE
5:     **while** $\max_{x \in M_t} \beta_{(i)}^{1/2}\sigma_t(x) \leq (1 + \overline{\delta})\eta_{(i)}$ **do**
6:         Increment $i$, set $\eta_{(i)} = r \times \eta_{(i-1)}$.

---

The choices of $\beta_{(i)}$, $\overline{\delta}$, and $r$ are discussed in Section 4. As with previous works, the kernel is assumed known in our theoretical results, whereas in practice it is typically learned from training data [3]. Characterizing the effect of model mismatch or online hyperparameter updates is beyond the scope of this paper, but is an interesting direction for future work.

**Algorithm 2** Parameter Updates for TRUVAR

**Input:** Selected points and observations $\{x_{t'}\}_{t'=1}^{t}$; $\{y_{t'}\}_{t'=1}^{t}$, previous sets $M_{t-1}$, $H_{t-1}$, $L_{t-1}$, parameter $\beta_{(i)}^{1/2}$, and for LSE, level-set threshold $h$.

1: Update $\mu_t$ and $\sigma_t$ according to (1)–(2), and form the upper and lower confidence bounds

$$u_t(x) = \mu_t(x) + \beta_{(i)}^{1/2}\sigma_t(x), \quad \ell_t(x) = \mu_t(x) - \beta_{(i)}^{1/2}\sigma_t(x). \tag{4}$$

2: For BO, set

$$M_t = \left\{ x \in M_{t-1} \: : \: u_t(x) \geq \max_{\overline{x} \in M_{t-1}} \ell_t(\overline{x}) \right\}, \tag{5}$$

or for LSE, set

$$M_t = \left\{ x \in M_{t-1} \: : \: u_t(x) \geq h \text{ and } \ell_t(x) \leq h \right\} \tag{6}$$

$$H_t = H_{t-1} \cup \left\{ x \in M_{t-1} \: : \: \ell_t(x) > h \right\}, \quad L_t = L_{t-1} \cup \left\{ x \in M_{t-1} \: : \: u_t(x) < h \right\}. \tag{7}$$

Some variants of our algorithm and theory are discussed in the supplementary material due to lack of space, including *pure* variance reduction, non-Bayesian settings [3], continuous domains [3], the batch setting [4], and implicit thresholds for level-set estimation [2].

## 3 Theoretical Bounds

In order to state our results for BO and LSE in a unified fashion, we define a notion of $\epsilon$-*accuracy* for the two settings. That is, we define this term differently in the two scenarios, but then we provide theorems that simultaneously apply to both. All proofs are given in the supplementary material.

**Definition 3.1.** *After time step $t$ of* TRUVAR, *we use the following terminology:*

- *For BO, the set $M_t$ is $\epsilon$-accurate if it contains all true maxima $x^* \in \arg\max_x f(x)$, and all of its points satisfy $f(x^*) - f(x) \leq \epsilon$.*

- *For LSE, the triplet $(M_t, H_t, L_t)$ is $\epsilon$-accurate if all points in $H_t$ satisfy $f(x) > h$, all points in $L_t$ satisfy $f(x) < h$, and all points in $M_t$ satisfy $|f(x) - h| \leq \frac{\epsilon}{2}$.*

*In both cases, the* cumulative cost *after time $t$ is defined as $C_t = \sum_{t'=1}^{t} c(x_{t'})$.*

We use $\frac{\epsilon}{2}$ in the LSE setting instead of $\epsilon$ since this creates a region of size $\epsilon$ where the function value lies, which is consistent with the BO setting. Our performance criterion for level-set estimation is slightly different from that of [2], but the two are closely related.

### 3.1 General Result

**Preliminary definitions:** Suppose that the $\{\beta_{(i)}\}$ are chosen to ensure valid confidence bounds, i.e., $\ell_t(x) \leq f(x) \leq u_t(x)$ with high probability; see Theorem 3.1 and its proof below for such choices. In this case, we have after the $i$-th epoch that all points are either already discarded (BO) or classified (LSE), or are known up to the confidence level $(1 + \overline{\delta})\eta_{(i)}$. For the points with such confidence, we have $u_t(x) - \ell_t(x) \leq 2(1 + \overline{\delta})\eta_{(i)}$, and hence

$$u_t(x) \leq \ell_t(x) + 2(1 + \overline{\delta})\eta_{(i)} \leq f(x) + 2(1 + \overline{\delta})\eta_{(i)}, \tag{8}$$

and similarly $\ell_t(x) \geq f(x) - 2(1 + \overline{\delta})\eta_{(i)}$. This means that all points other than those within a gap of width $4(1 + \overline{\delta})\eta_{(i)}$ must have been discarded or classified:

$$M_t \subseteq \left\{ x \: : \: f(x) \geq f(x^*) - 4(1 + \overline{\delta})\eta_{(i)} \right\} =: \overline{M}_{(i)} \quad \text{(BO)} \tag{9}$$

$$M_t \subseteq \left\{ x \: : \: |f(x) - h| \leq 2(1 + \overline{\delta})\eta_{(i)} \right\} =: \overline{M}_{(i)} \quad \text{(LSE)} \tag{10}$$

Since no points are discarded or classified initially, we define $\overline{M}_{(0)} = D$.

For a collection of points $S = (x'_1, \ldots, x'_{|S|})$, possibly containing duplicates, we write the total cost as $c(S) = \sum_{i=1}^{|S|} c(x'_i)$. Moreover, we denote the posterior variance upon observing the points up to time $t - 1$ *and* the additional points in $S$ by $\sigma_{t-1|S}(\overline{x})$. Therefore, $c(x) = c(\{x\})$ and $\sigma_{t-1|x}(\overline{x}) = \sigma_{t-1|\{x\}}(\overline{x})$. The minimum cost (respectively, maximum cost) is denoted by $c_{\min} = \min_{x \in D} c(x)$ (respectively, $c_{\max} = \max_{x \in D} c(x)$).

Finally, we introduce the quantity

$$C^*(\xi, M) = \min_S \left\{ c(S) \ : \ \max_{\overline{x} \in M} \sigma_{0|S}(\overline{x}) \leq \xi \right\}, \tag{11}$$

representing the minimum cost to achieve a posterior standard deviation of at most $\xi$ within $M$.

**Main result:** In all of our results, we make the following assumption.

**Assumption 3.1.** *The kernel $k(x, x')$ is such that the variance reduction function*

$$\psi_{t,x}(S) = \sigma_t^2(x) - \sigma_{t|S}^2(x) \tag{12}$$

*is submodular [17] for any time $t$, and any selected points $(x_1, \ldots, x_t)$ and query point $x$.*

This assumption has been used in several previous works based on Gaussian processes, and sufficient conditions for its validity can be found in [18, Sec. 8]. We now state the following general guarantee.

**Theorem 3.1.** *Fix $\epsilon > 0$ and $\delta \in (0, 1)$, and suppose there exist values $\{C_{(i)}\}$ and $\{\beta_{(i)}\}$ such that*

$$C_{(i)} \geq C^* \left( \frac{\eta_{(i)}}{\beta_{(i)}^{1/2}}, \overline{M}_{(i-1)} \right) \log \frac{|\overline{M}_{(i-1)}|\beta_{(i)}}{\overline{\delta}^2 \eta_{(i)}^2} + c_{\max}, \tag{13}$$

*and*

$$\beta_{(i)} \geq 2 \log \frac{|D|\left( \sum_{i' \leq i} C_{(i')} \right)^2 \pi^2}{6 \delta c_{\min}^2}. \tag{14}$$

*Then if* TRUVAR *is run with these choices of $\beta_{(i)}$ until the cumulative cost reaches*

$$C_\epsilon = \sum_{i \,:\, 4(1+\overline{\delta})\eta_{(i-1)} > \epsilon} C_{(i)}, \tag{15}$$

*then with probability at least $1 - \delta$, we have $\epsilon$-accuracy.*

While this theorem is somewhat abstract, it captures the fact that the algorithm improves when points having a lower cost and/or lower noise are available, since both of these lead to a smaller value of $C^*(\xi, M)$; the former by directly incurring a smaller cost, and the latter by shrinking the variance more rapidly. Below, we apply this result to some important cases.

### 3.2 Results for Specific Settings

**Homoscedastic and unit-cost setting:** Define the maximum mutual information [3]

$$\gamma_T = \max_{x_1,\ldots,x_T} \frac{1}{2} \log \det \left( \mathbf{I}_T + \sigma^{-2}\mathbf{K}_T \right), \tag{16}$$

and consider the case that $\sigma^2(x) = \sigma^2$ and $c(x) = 1$. In the supplementary material, we provide a theorem with a condition for $\epsilon$-accuracy of the form $T \geq \Omega^*\left( \frac{C_1 \gamma_T \beta_T}{\epsilon^2} + 1 \right)$ with $C_1 = \frac{1}{\log(1+\sigma^{-2})}$, thus matching [2, 4] up to logarithmic factors. In the following, we present a refined version that has a significantly better dependence on the noise level, thus exemplifying that a more careful analysis of (13) can provide improvements over the standard bounding techniques.

**Corollary 3.1.** *Fix $\epsilon > 0$ and $\delta \in (0, 1)$, define $\beta_T = 2 \log \frac{|D|T^2\pi^2}{6\delta}$, and set $\eta_{(1)} = 1$ and $r = \frac{1}{2}$. There exist choices of $\beta_{(i)}$ (not depending on the time horizon $T$) such that we have $\epsilon$-accuracy with probability at least $1 - \delta$ once the following condition holds:*

$$T \geq \left( 2\sigma^2\gamma_T\beta_T \frac{96(1+\overline{\delta})^2}{\epsilon^2} + C_1\gamma_T\beta_T \frac{6(1+\overline{\delta})^2}{\sigma^2} + 2\left\lceil \log_2 \frac{32(1+\overline{\delta})^2}{\epsilon\sigma} \right\rceil \right) \log \frac{16(1+\overline{\delta})^2|D|\beta_T}{\overline{\delta}^2\epsilon^2}, \tag{17}$$

*where $C_1 = \frac{1}{\log(1+\sigma^{-2})}$. This condition is of the form $T \geq \Omega^*\left( \frac{\sigma^2\gamma_T\beta_T}{\epsilon^2} + \frac{C_1\gamma_T\beta_T}{\sigma^2} + 1 \right)$.*

The choices $\eta_{(1)} = 1$ and $r = \frac{1}{2}$ are made for mathematical convenience, and a similar result follows for any other choices $\eta_{(1)} > 0$ and $r \in (0, 1)$, possibly with different constant factors.

As $\sigma^2 \to \infty$ (i.e., high noise), both of the above-mentioned bounds have noise dependence $O^*(\sigma^2)$, since $\log(1 + \alpha^{-1}) = O(\alpha^{-1})$ as $\alpha \to \infty$. On the other hand, as $\sigma^2 \to 0$ (i.e., low noise), $C_1$ is logarithmic, and Corollary 3.1 is significantly better provided that $\epsilon \ll \sigma$.

**Choosing the noise and cost:** Here we consider the setting that there is a domain of points $D_0$ that the reward function depends on, and alongside each point we can *choose* a noise variance $\sigma^2(k)$ $(k = 1, \ldots, K)$. Hence, $D = D_0 \times \{1, \cdots, K\}$. Lower noise variances incur a higher cost according to a cost function $c(k)$.

**Corollary 3.2.** *For each $k = 1, \cdots, K$, let $T^*(k)$ denote the smallest value of $T$ such that* (17) *holds with $\sigma^2(k)$ in place of $\sigma^2$, and with $\beta_T = 2 \log \frac{|D|T^2 c_{\max}^2 \pi^2}{6 \delta c_{\min}^2}$. Then, under the preceding setting, there exist choices of $\beta_{(i)}$ (not depending on $T$) such that we have $\epsilon$-accuracy with probability at least $1 - \delta$ once the cumulative cost reaches $\min_k c(k) T^*(k)$.*

This result roughly states that we obtain a bound as good as that obtained by sticking to *any fixed choice of noise level*. In other words, every choice of noise (and corresponding cost) corresponds to a different version of a BO or LSE algorithm (e.g., [2, 4]), and our algorithm has a similar performance guarantee to the best among all of those. This is potentially useful in avoiding the need for running an algorithm once per noise level and then choosing the best-performing one. Moreover, we found numerically that beyond matching the best fixed noise strategy, we can strictly improve over it by mixing the noise levels; see Section 4.

## 4 Experimental Results

We evaluate our algorithm in both the level-set estimation and Bayesian optimization settings.

**Parameter choices:** As with previous GP-based algorithms that use confidence bounds, our theoretical choice of $\beta_{(i)}$ in TRUVAR is typically overly conservative. Therefore, instead of using (14) directly, we use a more aggressive variant with similar dependence on the domain size and time: $\beta_{(i)} = a \log(|D| t_{(i)}^2)$, where $t_{(i)}$ is the time at which the epoch starts, and $a$ is a constant. Instead of the choice $a = 2$ dictated by (14), we set $a = 0.5$ for BO to avoid over-exploration. We found exploration to be slightly more beneficial for LSE, and hence set $a = 1$ for this setting. We found TRUVAR to be quite robust with respect to the choices of the remaining parameters, and simply set $\eta_{(1)} = 1$, $r = 0.1$, and $\bar{\delta} = 0$ in all experiments; while our theory assumes $\bar{\delta} > 0$, in practice there is negligible difference between choosing zero and a small positive value.

**Level-set estimation:** For the LSE experiments, we use a common classification rule in all algorithms, classifying the points according to the posterior mean as $\hat{H}_t = \{x : \mu_t(x) \geq h\}$ and $\hat{L}_t = \{x : \mu_t(x) < h\}$. The classification accuracy is measured by the $F_1$-score (i.e., the harmonic mean of precision and recall) with respect to the true super- and sub-level sets.

We compare TRUVAR against the GP-based LSE algorithm [2], which we name via the authors' surnames as GCHK, as well as the state-of-the-art straddle (STR) heuristic [7] and the maximum variance rule (VAR) [2]. Descriptions can be found in the supplementary material. GCHK includes an exploration constant $\beta_t$, and we follow the recommendation in [2] of setting $\beta_t^{1/2} = 3$.

**Lake data (unit cost):** We begin with a data set from the domain of environmental monitoring of inland waters, consisting of 2024 in situ measurements of chlorophyll concentration within a vertical transect plane, collected by an autonomous surface vessel in Lake Zürich [19]. As in [2], our goal is to detect regions of high concentration. We evaluate each algorithm on a $50 \times 50$ grid of points, with the corresponding values coming from the GP posterior that was derived using the original data (see Figure 2d). We use the Matérn-5/2 ARD kernel, setting its hyperparameters by maximizing the likelihood on the second (smaller) available dataset. The level-set threshold $h$ is set to $1.5$.

In Figure 2a, we show the performance of the algorithms averaged over 100 different runs; here the randomness is only with respect to the starting point, as we are in the noiseless setting. We observe that in this unit-cost case, TRUVAR performs similarly to GCHK and STR. All three methods outperform VAR, which is good for global exploration but less suited to level-set estimation.

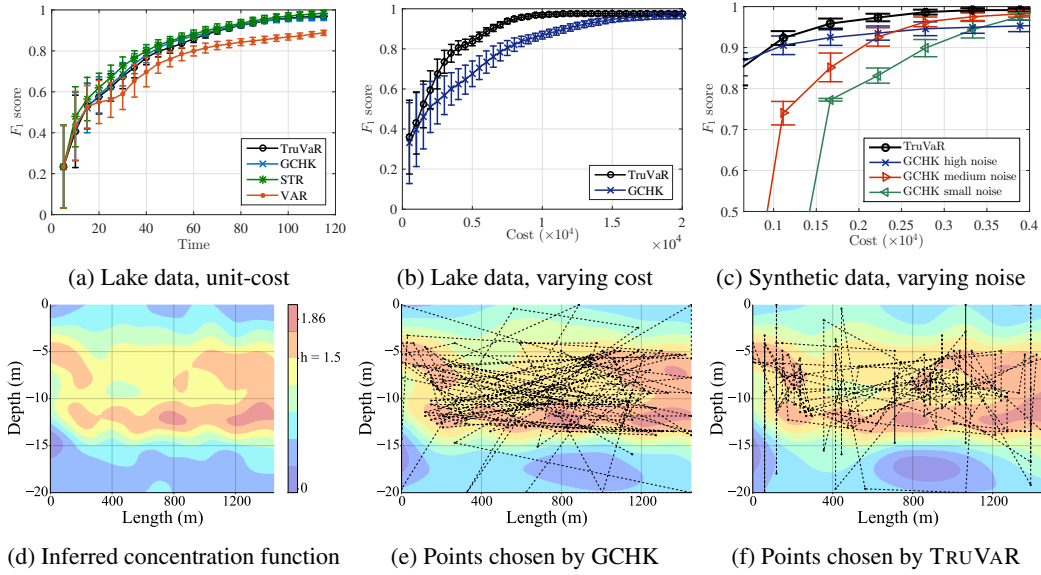

(a) Lake data, unit-cost     (b) Lake data, varying cost     (c) Synthetic data, varying noise

(d) Inferred concentration function     (e) Points chosen by GCHK     (f) Points chosen by TRUVAR

Figure 2: Experimental results for level-set estimation.

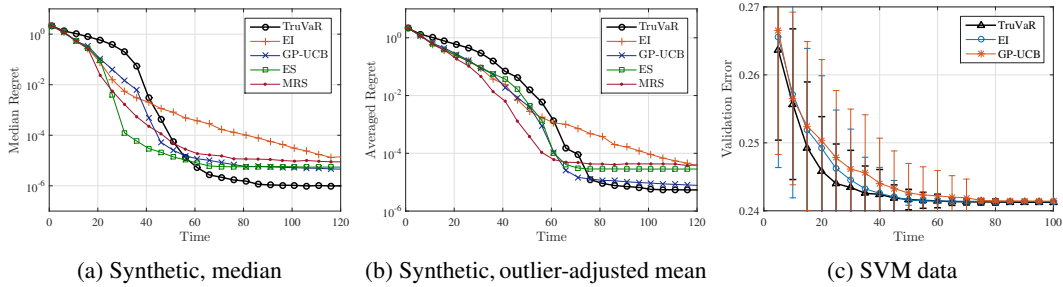

(a) Synthetic, median     (b) Synthetic, outlier-adjusted mean     (c) SVM data

Figure 3: Experimental results for Bayesian optimization.

**Lake data (varying cost):** Next, we modify the above setting by introducing pointwise costs that are a function of the previous sampled point $x'$, namely, $c_{x'}(x) = 0.25|x_1 - x'_1| + 4(|x_2| + 1)$, where $x_1$ is the vessel position and $x_2$ is the depth. Although we did not permit such a dependence on $x'$ in our original setup, the algorithm itself remains unchanged. Our choice of cost penalizes the distance traveled $|x_1 - x'_1|$, as well as the depth of the measurement $|x_2|$. Since incorporating costs into existing algorithms is non-trivial, we only compare against the original version of GCHK that ignores costs.

In Figure 2b, we see that TruVaR significantly outperforms GCHK, achieving a higher $F_1$ score for a significantly smaller cost. The intuition behind this can be seen in Figures 2e and 2f, where we show the points sampled by TruVaR and GCHK in one experiment run, connecting all pairs of consecutive points. GCHK is designed to pick few points, but since it ignores costs, the distance traveled is large. In contrast, by incorporating costs, TRUVAR tends to travel small distances, often even staying in the same $x_1$ location to take measurements at multiple depths $x_2$.

**Synthetic data with multiple noise levels:** In this experiment, we demonstrate Corollary 3.2 by considering the setting in which the algorithm can choose the sampling noise variance and incur the associated cost. We use a synthetic function sampled from a GP on a $50 \times 50$ grid with an isotropic squared exponential kernel having length scale $l = 0.1$ and unit variance, and set $h = 2.25$. We use three different noise levels, $\sigma^2 \in \{10^{-6}, 10^{-3}, 0.05\}$, with corresponding costs $\{15, 10, 2\}$.

We run GCHK separately for each of the three noise levels, while running TRUVAR as normal and allowing it to mix between the noise levels. The resulting $F_1$-scores are shown in Figure 2c. The best-performing version of GCHK changes throughout the time horizon, while TRUVAR is consistently better than all three. A discussion on how TRUVAR mixes between the noise levels can be found in the supplementary material.

**Bayesian optimization.** We now provide the results of two experiments for the BO setting.

**Synthetic data:** We first conduct a similar experiment as that in [8, 11], generating 200 different test functions defined on $[0,1]^2$. To generate a single test function, 200 points are chosen uniformly at random from $[0,1]^2$, their function values are generated from a GP using an isotropic squared exponential kernel with length scale $l = 0.1$ and unit variance, and the resulting posterior mean forms the function on the whole domain $[0,1]^2$. We subsequently assume that samples of this function are corrupted by Gaussian noise with $\sigma^2 = 10^{-6}$. The extension of TRUVAR to continuous domains is straightforward, and is explained in the supplementary material. For all algorithms considered, we evaluate the performance according to the regret of a single reported point, namely, the one having the highest posterior mean.

We compare the performance of TRUVAR against expected improvement (EI), GP-upper confidence bound (GP-UCB), entropy search (ES) and minimum regret search (MRS), whose acquisition functions are outlined in the supplementary material. We use publicly available code for ES and MRS [20]. The exploration parameter $\beta_t$ in GP-UCB is set according to the recommendation in [3] of dividing the theoretical value by five, and the parameters for ES and MRS are set according to the recommendations given in [11, Section 5.1].

Figure 3a plots the median of the regret, and Figure 3b plots the mean after removing outliers (i.e., the best and worst 5% of the runs). In the earlier rounds, ES and MRS provide the best performance, while TRUVAR improves slowly due to exploration. However, the regret of TRUVAR subsequently drops rapidly, giving the best performance in the later rounds after "zooming in" towards the maximum. GP-UCB generally performs well with the aggressive choice of $\beta_t$, despite previous works' experiments revealing it to perform poorly with the theoretical value.

**Hyperparameter tuning data:** In this experiment, we use the *SVM on grid* dataset, previously used in [21]. A $25 \times 14 \times 4$ grid of hyperparameter configurations resulting in $1400$ data points was pre-evaluated, forming the search space. The goal is to find a configuration with small validation error. We use a Matérn-5/2 ARD kernel, and re-learn its hyperparameters by maximizing the likelihood after sampling every 3 points. Since the hyperparameters are not fixed in advance, we replace $M_{t-1}$ by $D$ in (5) to avoid incorrectly ruling points out early on, allowing some removed points to be added again in later steps. Once the estimated hyperparameters stop to vary significantly, the size of the set of potential maximizers decreases almost monotonically. Since we consider the noiseless setting here, we measure performance using the simple regret, i.e., the best point found so far.

We again average over $100$ random starting points, and plot the resulting validation error in Figure 3c. Even in this noiseless and unit-cost setting that EI and GP-UCB are suited to, we find that TRUVAR performs slightly better, giving a better validation error with smaller error bars.

## 5 Conclusion

We highlight the following aspects in which TRUVAR is versatile:

- **Unified optimization and level-set estimation:** These are typically treated separately, whereas TRUVAR and its theoretical guarantees are essentially identical in both cases
- **Actions with costs:** TRUVAR naturally favors cost-effective points, as this is directly incorporated into the acquisition function.
- **Heteroscedastic noise:** TRUVAR chooses points that effectively shrink the variance of *other* points, thus directly taking advantage of situations in which some points are noisier than others.
- **Choosing the noise level:** We provided novel theoretical guarantees for the case that the algorithm can choose both a point and a noise level, *cf.*, Corollary 3.2.

Hence, TRUVAR directly handles several important aspects that are non-trivial to incorporate into myopic algorithms. Moreover, compared to other BO algorithms that perform a lookahead (e.g., ES and MRS), TRUVAR avoids the computationally expensive task of averaging over the posterior and/or measurements, and comes with rigorous theoretical guarantees.

**Acknowledgment:** This work was supported in part by the European Commission under Grant ERC Future Proof, SNF Sinergia project CRSII2-147633, SNF 200021-146750, and EPFL Fellows Horizon2020 grant 665667.

## Footnotes

[1]Extensions to continuous domains are discussed in the supplementary material.

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
