[Supplementary Material]

# Supplementary Material

**Truncated Variance Reduction: A Unified Approach to Bayesian Optimization and Level-Set Estimation** (Ilija Bogunovic, Jonathan Scarlett, Andreas Krause, and Volkan Cevher, NIPS 2016)

## A    Variations of the TRUVAR Algorithm

Our algorithm TRUVAR can naturally be adapted to suit various settings, including the following:

- **Non-monotonic $M_t$:** We have defined our sets $M_t$ to become smaller on every time step. However, if $\beta_{(i)}$ is chosen aggressively, it may be preferable to replace $M_{t-1}$ by $D$ in (5)–(6), in which case some removed points may be added back in depending on how the posterior mean changes between steps. We take this approach in the real-world BO example of Section 4 in which the kernel hyperparameters are learned online, so as to avoid incorrectly ruling out points early on due to mismatched hyperparameters.

- **Avoiding computing the acquisition function everywhere:** We found that instead of computing the acquisition function at every point in $D$, limiting the selection to points in $M_{t-1}$ has minimal effect on the performance. To reduce the computation even further, one could adopt a strategy such as that proposed in [22]: Take some relatively small number of points having the top GP-UCB or EI score, and then choose the point *in that restricted subset* having the highest score according to (3). In fact, the numerical results in Figure 3 suggest that this may not only reduce the computation, but also improve the performance in the very early rounds by making the algorithm *initially* behave more like GP-UCB or EI.

- **Pure variance reduction:** Setting $\eta_{(1)} = 0$ yields a *pure variance reduction* algorithm, which minimizes the total variance within $M_t$ via a one-step lookahead. While our theory does not apply in this case, we found this choice to also work well in practice.

- **Implicit threshold for level-set estimation:** While we have focused on a threshold $h$ for level-set estimation that is fixed in advance, one can easily incorporate the ideas of [2] to allow for an *implicit* threshold which is equal to some constant multiple of the function's maximum, which is random and unknown in advance.

- **Anticipating changes in $M_t$:** The acquisition function (3) computes the truncated variance reduction resulting from a one-step lookahead, but still sums over the previous set $M_{t-1}$. In order to make it more preferable to choose points that shrink $M_t$ faster, it may be preferable to instead sum over $M_{t-1|x}$, defined to be the updated set upon adding $x$. The problem here is that such an update depend on the next posterior mean, whose update requires sampling $f$. One solution is to average over the measurement as in [8]; alternatively, a simpler approach is to replace the observed value with its mean when doing this one-step lookahead, and then using the true observed function sample only when $x_t$ is actually chosen.

- **Continuous domains:** Our algorithm also extends to compact domains such as $[0,1]^d$. The main challenge is that the summations in (3) become integrals that need to be approximated numerically. The simplest way of doing this is to approximate the integrals by summations over a finite number of *representer points*, e.g., a grid of values that cover the domain sufficiently densely. The theoretical analysis of this modified algorithm is left for future work.

- **Batch setting:** As we show in the proof of Theorem 3.1, our algorithm can be interpreted as performing the first step of a greedy submodular covering problem at each time step. This leads to a very natural extension to the batch setting, in which multiple points (say, $k$ of them) are chosen at each time step: Simply perform the first $k$ steps of the greedy covering algorithm during each batch.

## B    Further Details of Numerical Experiments

**Other algorithms considered:** We outline the algorithms that TRUVAR is compared against; full details can be found in the cited papers. For level-set estimation, we have the following:

(a)                                                          (b)

Figure 4: (a) Function used in synthetic level-set estimation experiments; (b) The total cost used by TRUVAR for each of the three noise levels.

- The GCHK algorithm [2] evaluates, at each iteration, the point that is not yet classified with the largest ambiguity: $x_t = \arg\max_{x \in M_{t-1}} \min\{u_t(x) - h, h - \ell_t(x)\}$, where $u_t$ and $\ell_t$ are defined as in (4) with a parameter $\beta_t$ replacing $\beta_{(i)}$. Here, similarly to our algorithm, $M_{t-1}$ is the set of points that have not yet been classified as having a value above or below the threshold $h$.

- The straddle (STR) heuristic [7] chooses $x_t = \arg\max_{x \in D} 1.96\sigma_{t-1}(x) - |\mu_{t-1}(x) - h|$, favoring high-uncertainty points that are expected to have function values closer to $h$.

- The maximum variance rule (VAR) [2] simply chooses $x_t = \arg\max_{x \in D} \sigma_{t-1}(x)$.

For Bayesian optimization, we have the following:

- The expected improvement (EI) algorithm [1] chooses $x_t = \arg\max_{x \in D} \mathbb{E}_t[(f(x) - \xi_t)\mathbb{1}\{f(x) > \xi_t\}]$, where $\mathbb{E}_t[\cdot]$ denotes averaging with respect to the posterior distribution at time $t$, and $\xi_t$ is the best observed value so far. Since the posterior is Gaussian, the expectation can easily be expressed in closed form.

- The Gaussian Process Upper Confidence Bound (GP-UCB) algorithm [3] chooses the points with the highest upper confidence bounds, $x_t = \arg\max_{x \in D} \mu_{t-1}(x) + \beta_t \sigma_{t-1}(x)$, where $\beta_t$ is a parameter controlling the level of exploration performed.

- The Entropy Search (ES) algorithm can be interpreted as approximating the rule $x_t = \arg\min_{x \in D} h(f_{x^*|t,x})$, where $h(f) = \int_D f(x) \log \frac{1}{f(x)} dx$ denotes the differential entropy, and $f_{x^*|t,x}$ denotes the density function of the optimal action $x^*$ given the observations up to time $t$ along with the additional observation $x$. Intuitively, this rule seeks to minimize the uncertainty of $x^*$. Since its exact evaluation is intractable, it is approximated using Monte Carlo techniques to average with respect to the posterior distribution and the measurements.

- The Minimum Regret Search (MRS) algorithm [11] also resembles ES, but works with the expected regret instead of the differential entropy. Once again, Monte Carlo techniques are used to average with respect to the posterior distribution and the measurements.

**Efficiently computing the acquisition function:** To compute the value of the acquisition function (3) for different $x \in D$, we need to compute $\sigma^2_{t-1|x}(M_{t-1}) \in \mathbb{R}^{|M_{t-1}|}$, i.e., the posterior variance of points in $M_{t-1}$ upon observing $x$ along with $x_1, \cdots, x_{t-1}$. Instead of computing it directly, it is more efficient to recursively compute $\sigma^2_{t-1|x}(M_{t-1}) = \sigma^2_{t-1}(M_{t-1}) - \Delta_{t-1|x}(M_{t-1})$. The difference term, $\Delta_{t-1|x}(M_{t-1})$, can be computed as [8]:

$$\Delta_{t-1|x}(M_{t-1}) = \mathrm{diag}\big(\mathrm{Cov}_{t-1}(M_{t-1}, x)(\sigma^2 + \sigma^2_{t-1}(x))^{-1}\mathrm{Cov}_{t-1}(M_{t-1}, x)^T\big), \qquad (18)$$

where

$$\sigma^2_{t-1}(x) = k(x,x) - \mathbf{k}_{t-1}(x)^T\big(\mathbf{K}_{t-1} + \mathbf{\Sigma}_{t-1}\big)^{-1}\mathbf{k}_{t-1}(x) \qquad (19)$$

$$\mathrm{Cov}_{t-1}(M_{t-1}, x) = \mathbf{k}(M_{t-1}, x) - \mathbf{k}_{t-1}(M_{t-1})^T\big(\mathbf{K}_{t-1} + \mathbf{\Sigma}_{t-1}\big)^{-1}\mathbf{k}_{t-1}(x), \qquad (20)$$

and where $\mathbf{k}(M_{t-1}, x) = [k(\overline{x}, x)]_{\overline{x} \in |M_{t-1}|} \in \mathbb{R}^{|M_{t-1}|}$, and $\mathbf{k}_{t-1}(M_{t-1}) = [k(\overline{x}, x)]_{\overline{x} \in |M_{t-1}|, x \in \{1,\ldots,t-1\}} \in \mathbb{R}^{|M_{t-1}| \times (t-1)}$. When the Cholesky decomposition of $\mathbf{K}_{t-1} + \mathbf{\Sigma}_{t-1}$ is known, $\big(\mathbf{K}_{t-1} + \mathbf{\Sigma}_{t-1}\big)^{-1}\mathbf{k}_{t-1}(x)$ can be computed in time $O(t^2)$.

**Details on LSE experiment with multiple noise levels:** Figure 4a plots the randomly-generated function that was used in this experiment. Figure 4b plots the average cost spent by TRUVAR on each noise level by the end of the experiment, again averaged over 100 trials. We see that the cost is roughly equally distributed across the three levels. To be more specific, we observed that TRUVAR initially chooses high noise levels in order to cheaply explore, and throughout the course of the experiments, it gradually switches to lower noise levels in order to accurately determine the function values around the maximum. This is consistent with the behavior of the three version of GCHK, with $\sigma^2 = 0.05$ performing well in the early stages, but $\sigma^2 = 10^{-6}$ being preferable in the later stages.

**Extension of TRUVAR for synthetic BO experiment** We used the extension of TRUVAR to continuous domains outlined in Appendix A, approximating the integrals over $M_t$ by summations that are restricted to points on a uniformly-spaced $50 \times 50$ grid covering $[0,1]^2$. We optimized our acquisition function using DIRECT [23].

## C Proof of General Result (Theorem 3.1)

We begin with the following lemma from [3].

**Lemma C.1.** [3] *For each $t$, define $\beta_t = 2 \log \frac{|D| t^2 \pi^2}{6\delta}$. With probability at least $1 - \delta$, we have for all $x$ and $t$ that $|f(x) - \mu_t(x)| \le \beta_t^{1/2} \sigma_t(x)$.*

We conclude that in order for $\mu_t(\cdot) \pm \beta_{(i)}^{1/2} \sigma_t(\cdot)$ to provide valid confidence bounds, it suffices to ensure that $\beta_{(i)} \ge \beta_t$ for all $t$ in epoch $i$. From (14), we see that this is true provided that the time taken to reach the end of the $i$-th epoch is at most $\frac{1}{c_{\min}} \sum_{i' \le i} C_{(i')}$. Since $c_{\min}$ is the minimum pointwise cost, this holds provided that the cost incurred in epoch $i$ is at most $C_{(i)}$. The bulk of the proof is devoted to showing that this is the case.

We connect TRUVAR with the following budgeted submodular covering problem:[2]

$$\text{minimize}_S \ c(S) \quad \text{subject to } g_t(S) = g_{t,\max}, \tag{21}$$

where

$$g_t(S) = \sum_{\overline{x} \in M_{t-1}} \max \left\{ \sigma_{t-1}^2(\overline{x}), \frac{\eta_{(i)}^2}{\beta_{(i)}} \right\} - \sum_{\overline{x} \in M_{t-1}} \max \left\{ \sigma_{t-1|S}^2(\overline{x}), \frac{\eta_{(i)}^2}{\beta_{(i)}} \right\}, \tag{22}$$

and where $g_{t,\max}$ is the highest possible value of $g_t(S)$ over arbitrarily large $S$, i.e., it is the value obtained once all of the summands in the second summation in (22) have saturated to $\frac{\eta_{(i)}^2}{\beta_{(i)}}$:

$$g_{t,\max} = \sum_{\overline{x} \in M_{t-1}} \left( \max \left\{ \sigma_{t-1}^2(\overline{x}), \frac{\eta_{(i)}^2}{\beta_{(i)}} \right\} - \frac{\eta_{(i)}^2}{\beta_{(i)}} \right) \tag{23}$$

$$= \sum_{\overline{x} \in M_{t-1}} \max \left\{ 0, \sigma_{t-1}^2(\overline{x}) - \frac{\eta_{(i)}^2}{\beta_{(i)}} \right\}. \tag{24}$$

We thus refer to $g_{t,\max}$ as the *excess variance*; see Figure 5 for an illustration. Note that each time instant $t$ corresponds to a different function $g_t(S)$, and we are considering sets $S$ of an arbitrary size even though our algorithm only chooses one point at each time instant.

By our assumption on the submodularity of the variance reduction function, and the fact that taking the minimum with a constant[3] preserves submodularity [17], $g_t(S)$ is also submodular. It is also easily seen to be monotonically increasing, and normalized in the sense that $g_t(\emptyset) = 0$.

Figure 5: Illustration of the excess variance $g_{t,\max}$.

Our selection rule (3) at time $t$ can now be interpreted as the first step in a greedy algorithm for solving the budgeted submodular optimization problem (21); specifically, the greedy rule optimizes the objective value per unit cost. To obtain performance guarantees, we use Lemma 2 of [24] specialized to $|S| = 1$, which reads as follows in our own notation:

$$g_t(\{x_t\}) \geq \frac{c(x_t)}{c(S_t^*)} g_{t,\max}, \tag{25}$$

where $S_t^*$ is an optimal solution to (21), and hence $g_t(S^*) = g_{t,\max}$. Here $x_t$ is the point chosen greedily by our algorithm.

We now consider the behavior of the excess variance $g_{t,\max}$ in a single epoch, i.e., the duration of a single value of $i$ in the algorithm. We claim that for $t$ and $t+1$ in the same epoch, we have

$$g_{t+1,\max} \leq g_{t,\max} - g_t(\{x_t\}). \tag{26}$$

To see this, we note from (22)–(23) that this would hold with equality if we were to have $M_t = M_{t-1}$, since by definition we have $\sigma_t^2(\bar{x}) = \sigma_{t-1|\{x_t\}}^2(\bar{x})$. We therefore obtain (26) by recalling that $M_t$ is decreasing in $t$ with respect to inclusion, and noting from (24) that any given $g_{t,\max}$ can only decrease when $M_{t-1}$ is smaller.

Combining (25)–(26) gives

$$g_{t+1,\max} \leq \left(1 - \frac{c(x_t)}{c(S_t^*)}\right) g_{t,\max}. \tag{27}$$

We also note that $c(S_{t+1}^*) \leq c(S_t^*)$, which follows since $\sigma_t^2(\cdot)$ is decreasing in $t$ and $M_t$ is shrinking in $t$, and therefore at time $t+1$ a smaller cost is required to ensure that all terms in the second summation of (22) have saturated to $\frac{\eta_{(i)}^2}{\beta_{(i)}}$. Hence, and applying (27) recursively, we obtain for $t$ and $t+\ell$ in the same epoch that

$$\frac{g_{t+\ell,\max}}{g_{t,\max}} \leq \prod_{t'=t+1}^{t+\ell} \left(1 - \frac{c(x_{t'})}{c(S_t^*)}\right) \tag{28}$$

$$\leq \exp\left(-\frac{\sum_{t'=t+1}^{t+\ell} c(x_{t'})}{c(S_t^*)}\right), \tag{29}$$

where we have applied the inequality $1 - \alpha \leq e^{-\alpha}$. Moreover, the total cost incurred by choosing these points is precisely $\sum_{t'=t+1}^{t+\ell} c(x_t)$. Thus, letting $t_{(i)}$ be the first time index in the $i$-th epoch, we find that in order to remove all but a proportion $\gamma$ of the initial excess variance $g_{t_{(i)},\max}$, it suffices that the cost incurred is at least

$$c(S_{t_{(i)}}^*) \log \frac{1}{\gamma}. \tag{30}$$

Next, we observe that since the posterior variance is upper bounded by one due to the assumption $k(x, x) = 1$, the initial excess variance $g_{t_{(i)},\max}$ is upper bounded by $g_{t_{(i)},\max} \leq |M_{t_{(i)}-1}|$, the size

of the set of potential maximizers at the *start* of the epoch. It follows that if we set

$$\gamma = \frac{\overline{\delta}^2 \eta_{(i)}^2}{\beta_{(i)} |M_{t_{(i)}-1}|}, \tag{31}$$

then removing all but a proportion $\gamma$ of $g_{t_{(i)},\max}$ also implies removing all but $\overline{\delta}^2 \frac{\eta_{(i)}^2}{\beta_{(i)}}$ of it. In other words, if at time $t$ we have incurred a cost in epoch $i$ satisfying (30) with $\gamma$ as in (31), then we must have $g_{t,\max} \le \overline{\delta}^2 \frac{\eta_{(i)}^2}{\beta_{(i)}}$.

Removing all of the excess variance would imply $\eta_{(i)}$-confidence at all points in $M_t$. In the worst case, the remaining excess variance $\overline{\delta}^2 \frac{\eta_{(i)}^2}{\beta_{(i)}}$ is concentrated entirely on a single point, in which case its confidence is upper bounded by $\sqrt{1 + \overline{\delta}^2}\, \eta_{(i)}$, which is further upper bounded by $(1 + \overline{\delta})\eta_{(i)}$ due to the identity $\sqrt{1 + \alpha^2} \le 1 + \alpha$.

Combining these observations, we conclude that in the $i$-th epoch, upon incurring a cost of at least

$$c(S^*_{t_{(i)}}) \log \frac{|M_{t_{(i)}-1}| \beta_{(i)}}{\overline{\delta}^2 \eta_{(i)}^2}, \tag{32}$$

we are guaranteed to have $(1 + \overline{\delta})\eta_{(i)}$-confidence for all points in $M_t$. Having such confidence is precisely the condition used in the algorithm to move onto the next epoch, and we conclude that the epoch must end by (or sooner than) the time that (32) holds.

In accordance with the discussion following Lemma C.1, we need to show that when the high-probability event in that lemma holds true, (32) is upper bounded by the right-hand side of (13) for all epochs. We do this via an induction argument on the epoch number:

- As a base case, recalling that $M_0 = M_{(0)} = D$, we find that (32) and (13) coincide, with the addition of $c_{\max}$ arising since once (32) is exceeded, it may be exceeded by any amount up to $c_{\max}$.

- Fix an epoch number $i > 1$, and suppose that for all $i' < i$, the cost incurred in epoch $i'$ was at most $C^*\big(\frac{\eta_{(i')}}{\beta_{(i')}^{1/2}}, \overline{M}_{(i'-1)}\big) \log \frac{|\overline{M}_{(i'-1)}| \beta_{(i')}}{\overline{\delta}^2 \eta_{(i')}^2} + c_{\max}$. By the choice of $\beta_{(i')}$ in (14), we find that under the event in Lemma C.1, the confidence bounds $\mu_t \pm \sqrt{\beta_{(i')}}\sigma_t$ must have been valid for all $t$ in the epochs $i' < i$, and hence $\overline{M}_{(i-1)} \subseteq M_{t_{(i)}-1}$ (*cf.*, (9)–(10)).

  From this, we claim that an analogous argument to the base case implies that (32) is upper bounded by the right-hand side of (13), as required. The only additional argument compared to the base case is noting that $c(S^*_{t_{(i)}})$ defines the minimum cost to uniformly shrink the posterior standard deviation within $M_{t_{(i)}-1}$ down to $\frac{\eta_{(i)}^2}{\beta_{(i)}}$ after already having chosen $x_1, \ldots, x_{t_{(i)}-1}$, whereas $C^*\big(\frac{\eta_{(i)}}{\beta_{(i)}^{1/2}}, \overline{M}_{(i-1)}\big)$ is defined analogously for the set $\overline{M}_{(i-1)}$ with no previously-chosen points. The latter clearly upper bounds the former.

Finally, we check the conditions for $\epsilon$-accuracy in Definition 3.1. In the case of BO, summing (13) over all of the epochs such that $4(1 + \overline{\delta})\eta_{(i-1)} > \epsilon$ yields (15); recall from (9) that after any epoch $i$ such that $4(1 + \overline{\delta})\eta_{(i)} \le \epsilon$, all points are at most $\epsilon$-suboptimal. We also note that all true maxima must remain in $M_t$, due to the fact that we showed $\beta_{(i)}$ yields valid confidence bounds with high probability, and we only ever discard points that are deemed suboptimal according to those bounds. For LSE, a similar conclusion follows from (10) by summing over all epochs such that $2(1 + \overline{\delta})\eta_{(i-1)} > \frac{\epsilon}{2}$, which is the same as $4(1 + \overline{\delta})\eta_{(i-1)} > \epsilon$. Once again, all points in $H_t$ and $L_t$ are correct due to the validity of our confidence bounds.

# D  Simplified Result for the Homoscedastic and Unit-Cost Setting

Since we are focusing on unit costs $c(x) = 1$, the cost simply corresponds to the number of rounds $T$. To highlight this fact, we replace $C^*$ in (11) by

$$T^*(\xi, M) = \min_S \left\{ |S| \, : \, \max_{\overline{x} \in M} \sigma_{0|S}(\overline{x}) \leq \xi \right\}, \tag{33}$$

and similarly replace (34)–(36) by

$$T_{(i)} \geq T^*\left( \frac{\eta_{(i)}}{\beta_{(i)}^{1/2}}, \overline{M}_{(i-1)} \right) \log \frac{|\overline{M}_{(i-1)}| \beta_{(i)}}{\overline{\delta}^2 \eta_{(i)}^2} + 1 \tag{34}$$

$$\beta_{(i)} \geq 2 \log \frac{|\overline{M}_{(i-1)}| \left( \sum_{i' \leq i} T_{(i')} \right)^2 \pi^2}{6\delta} \tag{35}$$

$$T_\epsilon = \sum_{i \, : \, 4(1+\overline{\delta})\eta_{(i-1)} > \epsilon} T_{(i)}. \tag{36}$$

In this section, we prove the following as an application of Theorem 3.1.

**Corollary D.1.** *Fix $\epsilon > 0$ and $\delta \in (0,1)$, define $\beta_T = 2 \log \frac{|D|T^2\pi^2}{6\delta}$, and set $\eta_{(1)} = 1$ and $r = \frac{1}{2}$. There exist choices of $\beta_{(i)}$ (not depending on the time horizon $T$) such that we have $\epsilon$-accuracy with probability at least $1 - \delta$ once the following condition holds:*

$$T \geq \left( C_1 \gamma_T \beta_T \frac{96(1+\overline{\delta})^2}{\epsilon^2} + 2 \left\lceil \log_2 \frac{8(1+\overline{\delta})}{\epsilon} \right\rceil \right) \log \frac{16(1+\overline{\delta})^2 |D| \beta_T}{\overline{\delta}^2 \epsilon^2}, \tag{37}$$

*where $C_1 = \frac{1}{\log(1+\sigma^{-2})}$. This condition is of the form $T \geq \Omega^*\left( \frac{C_1 \gamma_T \beta_T}{\epsilon^2} + 1 \right)$.*

We bound the cardinality of $S$ in (33) by considering a procedure that greedily picks $\arg \max_{x \in M} \sigma_t(x)$. We claim that after selecting $k$ points according to this procedure to construct a set $S_k$, we have

$$\max_x \sigma_{0|S_k}^2(x) \leq C_1 \frac{\gamma_k}{k}, \tag{38}$$

where $C_1 = \frac{1}{\log(1+\sigma^{-2})}$. This is seen by writing

$$k \max_{x \in M} \sigma_{0|S_k}^2(x) = k \sigma_{0|S_k}^2(x_k) \tag{39}$$

$$\leq \sum_{j=1}^k \sigma_{0|S_j}^2(x_j) \tag{40}$$

$$\leq \frac{1}{\log(1+\sigma^{-2})} \gamma_k, \tag{41}$$

where we respectively used that $x_k$ maximizes $\sigma_{0|S_k}$, that $\sigma_{0|S_i}(x_i)$ always decreases as more points are chosen, and the bound on the sum of variances of sampled points from [3, Lemma 5.4].

Identifying $k$ with $T^*$, and $\max_{x \in M} \sigma_{0|S_k}(x)$ with $\xi = \frac{\eta}{\beta^{1/2}}$ (for some $\eta$ and $\beta$ to be specified), we obtain from (41) that

$$T^*\left( \frac{\eta}{\beta^{1/2}}, M \right) \leq \min \left\{ T^* \, : \, T^* \geq \frac{C_1 \gamma_{T^*} \beta}{\eta^2} \right\}. \tag{42}$$

Consider the value $T^*\left( \frac{\eta_{(i)}}{\beta_{(i)}^{1/2}}, \overline{M}_{(i-1)} \right)$ corresponding to the parameters $\eta = \eta_{(i)}$ and $\beta = \beta_{(i)}$ associated with epoch $i$. Letting $T = T_\epsilon$ denote the total time horizon, and using (36), we find that $\beta_{(i)}$ in (35) can be upper bounded by $2 \log \frac{|D|T^2\pi^2}{6\delta}$, which is precisely $\beta_T$. By similarly using the monotonicity of $\gamma_t$, we obtain

$$T^*\left( \frac{\eta_{(i)}}{\beta_{(i)}^{1/2}}, \overline{M}_{(i-1)} \right) \leq \frac{C_1 \gamma_T \beta_T}{\eta_{(i)}^2} + 1, \tag{43}$$

where the addition of one is to account for possible rounding up to the nearest integer.

Using (43), we find that in order for (34) to hold it suffices that

$$T_{(i)} \geq \left( \frac{C_1 \gamma_T \beta_T}{\eta_{(i)}^2} + 1 \right) \log \frac{|\overline{M}_{(i-1)}| \beta_{(i)}}{\overline{\delta}^2 \eta_{(i)}^2} + 1. \tag{44}$$

Since we are only considering values of $i$ such that $4(1 + \overline{\delta})\eta_{(i-1)} > \epsilon$, and since $\overline{M}_{(i-1)} \subseteq D$, we can upper bound the logarithm by $\log \frac{16(1+\overline{\delta})^2 |D| \beta_{(i)}}{\overline{\delta}^2 \epsilon^2} > 1$, and hence in order for (44) to hold it suffices that

$$T_{(i)} \geq \left( \frac{C_1 \gamma_T \beta_T}{\eta_{(i)}^2} + 2 \right) \log \frac{16(1 + \overline{\delta})^2 |D| \beta_T}{\overline{\delta}^2 \epsilon^2}. \tag{45}$$

We also note that since $\eta_{(i)} = \eta_{(1)} r^{i-1}$, the condition $4(1 + \overline{\delta})\eta_{(i-1)} > \epsilon$ is equivalent to

$$4(1 + \overline{\delta})\eta_{(1)} r^{i-2} > \epsilon \tag{46}$$

$$\iff r^{i-2} > \frac{\epsilon}{4(1 + \overline{\delta})\eta_{(1)}} \tag{47}$$

$$\iff i < 2 + \log_{1/r} \frac{4(1 + \overline{\delta})\eta_{(1)}}{\epsilon} \tag{48}$$

$$\iff i \leq \left\lceil \log_{1/r} \frac{4(1 + \overline{\delta})\eta_{(1)}}{\epsilon} \right\rceil + 1 \tag{49}$$

$$\iff i \leq \left\lceil \log_{1/r} \frac{4(1 + \overline{\delta})\eta_{(1)}}{r\epsilon} \right\rceil, \tag{50}$$

where in the last line we used $\log_{1/r} \frac{1}{r} = 1$. Summing (45) over all such $i$ in accordance with (36), we obtain following sufficient condition on the time horizon for $\epsilon$-accuracy:

$$T \geq \left( C_1 \gamma_T \beta_T \sum_{i=1}^{\lceil \log_{1/r} \frac{4(1+\overline{\delta})\eta_{(1)}}{\epsilon} \rceil + 1} \frac{1}{\eta_{(i)}^2} + 2 \left\lceil \log_{1/r} \frac{4(1 + \overline{\delta})\eta_{(1)}}{r\epsilon} \right\rceil \right) \log \frac{16(1 + \overline{\delta})^2 |D| \beta_T}{\overline{\delta}^2 \epsilon^2}. \tag{51}$$

Finally, we weaken this condition by upper bounding the summation as follows:

$$\sum_{i=1}^{\lceil \log_{1/r} \frac{4(1+\overline{\delta})\eta_{(1)}}{\epsilon} \rceil + 1} \frac{1}{\eta_{(i)}^2} = \sum_{i=1}^{\lceil \log_{1/r} \frac{4(1+\overline{\delta})\eta_{(1)}}{\epsilon} \rceil + 1} \frac{1}{\eta_{(1)}^2 r^{2(i-1)}} \tag{52}$$

$$= \sum_{i=0}^{\lceil \log_{1/r} \frac{4(1+\overline{\delta})\eta_{(1)}}{\epsilon} \rceil} \frac{1}{\eta_{(1)}^2 r^{2i}} \tag{53}$$

$$\leq \frac{1}{r^2(1 - r^2)} \frac{16(1 + \overline{\delta})^2}{\epsilon^2}, \tag{54}$$

where the last line follows from the identity $\sum_{i=0}^{\lceil \log_{1/r} A \rceil} \frac{1}{r^{2i}} \leq \frac{1}{r^2(1-r^2)} A^2$ for $\log_{1/r} A > 0$. Substituting $r = \frac{1}{2}$ and $\eta_{(1)} = 1$ concludes the proof; the former yields $\frac{1}{r^2(1-r^2)} = \frac{16}{3} \leq 6$.

## E   Proof of Improved Noise Dependence (Corollary 3.1))

The bound in (41) is based on the inequality [3, Lemma 5.4]

$$\frac{\sigma_t^2}{\sigma^2} \leq C_1 \log \left( 1 + \frac{\sigma_t^2}{\sigma^2} \right) \tag{55}$$

for $\sigma_t^2 \in [0, 1]$ (with $C_1 = \frac{\sigma^{-2}}{\log(1+\sigma^{-2})}$), which can be very loose when $\sigma^2$ is small. Our starting point to improve the noise dependence is to note that the following holds under the more restrictive

condition $\sigma_t^2 \le \sigma^2$:

$$\sigma_t^2 = \sigma^2 \frac{\sigma_t^2}{\sigma^2} \tag{56}$$

$$\le 2\sigma^2 \log\left(1 + \frac{\sigma_t^2}{\sigma^2}\right), \tag{57}$$

where we have used the fact that $\alpha \le 2\log(1+\alpha)$ for $\alpha \in [0,1]$.

The idea now is to use (57) in the epochs that are late enough so that $\sigma_t^2 \le \sigma^2$, and (41) in the earlier epochs. Since $(1+\bar{\delta})\eta_{(i)}$ is the confidence level obtained after epoch $i$, and since $\beta_{(i)}^{1/2}\sigma_t$ is the confidence level after time $t$, we find that in order to ensure $\sigma_t^2 \le \sigma^2$ it suffices that $\frac{(1+\bar{\delta})^2\eta_{(i)}^2}{\beta_{(i)}} \le \sigma^2$. Moreover, our choice of $\beta_{(i)}$ in (14) is always greater than one when $|D| \ge 2$ (which is a trivial condition), and hence we can weaken this condition to $(1+\bar{\delta})^2\eta_{(i)}^2 \le \sigma^2$, and write

$$\sum_i T_{(i)} \le \sum_{i\,:\,(1+\bar{\delta})^2\eta_{(i-1)}^2 > \sigma^2} T_{(i)}^{(C_1)} + \sum_i T_{(i)}^{(2\sigma^2)}, \tag{58}$$

where $T_{(i)}^{(C_1)}$ denotes bound on $T_{(i)}$ in (45) based on (41), and $T_{(i)}^{(C_1)}$ denotes the analogous bound based on (57) with $2\sigma^2$ in place of $C_1$. Similarly to (50), the first summation is over a subset of the range $i \le \lceil \log_{1/r} \frac{(1+\bar{\delta})\eta_{(1)}}{r\sigma} \rceil$, and it follows that the condition (51) may be replaced by

$$T \ge \left(2\sigma^2\gamma_T\beta_T \sum_{i=1}^{\lceil \log_{1/r} \frac{4(1+\bar{\delta})\eta_{(1)}}{\epsilon}\rceil + 1} \frac{1}{\eta_{(i)}^2} + 2\left\lceil \log_{1/r} \frac{4(1+\bar{\delta})\eta_{(1)}}{r\epsilon} \right\rceil\right) \log \frac{16(1+\bar{\delta})^2|D|\beta_T}{\bar{\delta}^2\epsilon^2}$$

$$+ \left(C_1\gamma_T\beta_T \sum_{i=1}^{\lceil \log_{1/r} \frac{(1+\bar{\delta})\eta_{(1)}}{\sigma}\rceil + 1} + 2\left\lceil \log_{1/r} \frac{(1+\bar{\delta})\eta_{(1)}}{r\sigma} \right\rceil\right) \log \frac{16(1+\bar{\delta})^2|D|\beta_T}{\bar{\delta}^2\epsilon^2}. \tag{59}$$

The first summation is handled in the same way as the previous subsection, and the second summation is upper bounded by writing

$$\sum_{i=1}^{\lceil \log_{1/r} \frac{(1+\bar{\delta})\eta_{(1)}}{\sigma}\rceil + 1} \frac{1}{\eta_{(i)}^2} = \sum_{i=1}^{\lceil \log_{1/r} \frac{(1+\bar{\delta})\eta_{(1)}}{\sigma}\rceil + 1} \frac{1}{\eta_{(1)}^2 r^{2(i-1)}} \tag{60}$$

$$\le \frac{(1+\bar{\delta})^2}{r^2(1-r^2)} \frac{1}{\sigma^2}, \tag{61}$$

where (60) follows since $\eta_{(i)} = \eta_{(1)} r^{i-1}$, and (61) follows in the same way as (54). Once again, setting $r = \frac{1}{2}$ and $\eta_{(1)} = 1$ concludes the proof, with the third term in (17) coming from the identity $2\lceil \log_2 \frac{8(1+\bar{\delta})}{\epsilon} \rceil + 2\lceil \log_2 \frac{2(1+\bar{\delta})}{\sigma} \rceil \le 2\lceil \log_2 \frac{32(1+\bar{\delta})^2}{\epsilon\sigma} \rceil$.

## F   Proof for the Setting of Choosing Noise (Corollary 3.2)

The proof follows the same arguments as those of Appendices D and E, with $C^*$ being upper bounded in $K$ different ways, one for each possible noise level. The choice $\beta_T = 2\log \frac{|D|T^2 c_{\max}^2 \pi^2}{6\delta c_{\min}^2}$ arises as a simple upper bound to the right-hand side of (14) resulting from the fact that $\sum_{t=1}^T c(x_t) \le c_{\max}T$.

## Footnotes

[2]Recall that $S$ may contain duplicates, and these are counted multiple times accordingly in the definitions of both $c(S)$ and $g_t(S)$. All of our equations can be cast in terms of standard sets (without duplicates) by expanding $D$ to $D \times \{1, \cdots, N\}$ for any integer $N$ that is larger than the maximum number of points that are chosen throughout the course of the algorithm.

[3]The minimum becomes a maximum after negation.