[Reviews · NeurIPS 2016]

Reviewer 1

Summary

The paper proposes unified framework for Bayesian optimisation and level set estimation. The authors do so by means of a new algorithm called TRUEVAR that shrinks the total surrogate model variance up to some threshold. The theoretical results of the paper are nicely stated and the experimental section is nice a very coherent with the rest of the work.

Qualitative Assessment

I quite like this paper and I think that it is a nice and incremental work in the field of Bayesian optimisation. The paper is very well structured and the idea of how Bayesian optimization and level set estimation are related is very well described. The theoretical results of the paper are interesting and the experiential section is convincing. I have no special objection to this work and I think if should be clearly accepted in the conference.

Confidence in this Review

3-Expert (read the paper in detail, know the area, quite certain of my opinion)


Reviewer 2

Summary

This manuscripts introduces a new algorithm called "truncated variance reduction" (TruVaR; I will write TV here) that is appropriate both for level set estimation and Bayesian optimization on nonlinear functions. The idea is to maintain a set of points that are either possible maximizers of the function (for Bayesian optimization) or points near the level set threshold (for LSE). We then greedily evaluate the function so as to minimize the total truncated variance of points in this set. When the maximum uncertainty in this set drops below the current truncation value, we decrease that value and continue. The authors provide a theoretical guarantee for the performance of this algorithm, providing a unified analysis for both LSE and Bayesian optimization in terms of a notion they call "epsilon-accuracy." The authors also consider the cases where the observation noise is heteroskedastic and when there is a known cost of observing a given point, providing simple extensions of the algorithm for these cases and proving a result in the nonuniform cost case. Finally, a series of experiments investigates the performance of the proposed method on the LSE and Bayesian optimization tasks.

Qualitative Assessment

This manuscript has some interesting ideas. The algorithm is simple, easy-to-understand, and the fact that it works is intuitive. The joint analysis of Bayesian optimization and LSE via "epsilon-accuracy" is nice, although it's not terribly surprising that this is possible. The analysis is somewhat straightforward and predictable, using many tools that are now common in algorithms like these (confidence bounds, maximum mutual information, submodularity). That said, the theoretical contributions certainly strengthen the proposed method. That said, I have a few complaints about this paper. The joint analysis of LSE and Bayesian optimization is interesting and perhaps even insightful. I think the LSE problem is interesting and deserving of more attention than it gets, which seems to at the rate of a paper every few years. Where this paper fails, however, is in purporting that these problems are of equal practical importance. This is not done so explicitly, but rather implicitly by certain choices made, e.g., in the experimental setup (see below). Although LSE is a compelling problem, and indeed has some applications, e.g., in environmental monitoring, it has a rather niche role in the scheme of things. This somewhat limits the potential impact of this paper. Bayesian optimization, on the other hand, has many natural applications across a range of domains. I say this not because Bayesian optimization is my pet problem (it's not), but rather out of simple pragmatism. A strong algorithmic contribution to Bayesian optimization can have a large impact indeed. Unfortunately, I don't think this paper does a sufficient job of selling the proposed algorithm to that audience, due to the experimental treatment. I believe the experiments are rather weak in comparison to the theoretical contributions here. LSE is investigated first, on an interesting real-world environmental monitoring dataset. The authors investigate the performance of TV against VAR (uncertainty sampling), a previous LSE algorithm, and the "straddle" algorithm from Bryan and Schneider, with TV appearing to work well but perhaps not quite as well as straddle. They turn their attention to the incorporation of cost (in this case, defined with respect to the movement of the autonomous vehicle), and show that TV outperforms LSE when measuring performance vs cost. This is not surprising due to the fact that LSE is agnostic of cost. The authors point out that their algorithm often causes the boat to remain stationary, instead choosing to measure at the same location but at a different depth. Even then, the boat moves around quite a bit along the transect. I can't help but ask how a simple grid design would perform, which could be realized with very little movement cost. The authors then turn their attention to Bayesian optimization, the more important of the two problems, and in my opinion give it short shrift experimentally. The test functions are synthetic draws from Gaussian processes and a single hyperparameter tuning experiment (where the results are inconclusive). I do not judge these to be sufficient to convince the Bayesian optimization community that "yet another acquisition function" is needed in an already crowded space. Varying costs of observation are also ignored, with the authors claiming that the LSE experiments "already demonstrated the effects of costs." I disagree. The LSE experiments show only that TV outperforms the cost-unaware LSE algorithm. Moreover, varying observation cost has received much attention in Bayesian optimization, and offering a better solution to that problem would significantly raise the impact of this contribution. Instead, the authors sweep it under the rug. A comparison against, e.g., EI / cost would go a long way here. I do not understand the sentence "Since the domain is discrete, we compare only against EI and GP-UCB." Entropy search is certainly tractable here, if only via sampling and rank-one updates. Is 24 * 14 * 4 (= 1344) a typo? 25 * 14 * 4 (= 1400)? Finally, the experiments are riddled with magic constants, some of which are not really motivated (although citations are given for some). For example, why the exact set of costs {15, 10, 2}? Why not some other set? Reducing the noise variance from 0.05 to 1e-6 is a 223-fold increase in fidelity. Why is that only 7.5 times more expensive? The manuscript is relatively well written and easy to read. However, many of the figures are difficult to read due to tiny text and/or small markers/details (e.g. Figure 1, Figures 2(e) and 2(f)). Reading the paper, I wondered whether there might be a nontrivial connection between TV and entropy search. It seems like TV will by construction be reducing the entropy of p(x* | D), if only indirectly. A bit of discussion on that might be interesting. For example, could it be possible that TV is approximately performing entropy search, using a cheap surrogate? If so, is it possible this could be made precise?

Confidence in this Review

3-Expert (read the paper in detail, know the area, quite certain of my opinion)


Reviewer 3

Summary

This paper combines the problems of Bayesian optimization and Level Set estimation in a single setup. The proposed algorithm is able to deal with both problems showing that information can be exploited from the synergy providing better results than treating the problem separately.

Qualitative Assessment

This paper was a pleasure to read. It is excellently written and the results are astonishing. The idea of dealing jointly the problems of BO and LSE and improving both results at the same time seems counter-intuitive. Both the theoretical contribution and the results are impressive. My only concerns have been mostly raised by the authors. First, the known kernel hyperparameters. The authors claim that the limitation is a requirement of the theoretical result, however, that restriction is propagated to the experiments, where the length scale is also fixed. Given the type of algorithms, it should be trivial to add ML or MCMC hyperparamenter estimation. Second, the constrain $x \in M_{t-1}$ seems reasonable, but I can imagine certain situations where it could be problematic for BO, specially due to initial model mismatch. Again, it would be interesting to know if that constrain is theoretical, practical or both.

Confidence in this Review

3-Expert (read the paper in detail, know the area, quite certain of my opinion)


Reviewer 4

Summary

The authors introduce truncated variance reduction as a unified approach to both Bayesian optimization and level-set estimation. They provide general guarantees for their algorithm covering these two areas and even strengthen existing results further. I enjoyed reading the paper a lot and recommend it to be accepted as a talk.

Qualitative Assessment

I like the paper a lot and don't have any major concerns about it. I have, however, some minor comments listed below by line. line 22: Looks like theres is a double-space before "We" line 32: just to name a few -> to name but a few line 68: I wouldn't say "vastly different" line 79: I would say something about how ES is set up for global optimization and not really a minimum-regret algorithm Figure 1 could need some more annotations. It is somewhat clear what it shows based on the description, but I would add more text into the plots. line 126: two tabs before "The algorithm"? line 158: wouldn't the truvar algorithm be even more negatively affected by a mis-specification of the kernel? line 194: unifies might be the wrong word here line 199: "and its proof such choices" ->missing of? line 201: known to be within? line 247: specialize->maybe apply to line 288: I really like that the algorithm can improve upon the fixed noise strategy as this makes it a nice alternative for many real world settings line 394: it would be interesting to see a mixed acquisition function that first explores and then uses entropy search, also another interesting thing to look at would be the minimization of regret over computational budget spent, so that one could really compare between ES and TruVar. line 397: what exactly does aggressive mean here? line 425: I'm not sure I'd make the multi-tak settings a bullet point as it wasn't shown in the paper. line 430: I get that it avoids that as compared to ES. Without taking a stance here, but one could argue that ES and TruVar might be set up for two different setups, global and Bayesian optimization? How does the algorithm compare to the safe exploration algorithm postulated by Krause et al.? Couldn't that also be incorporated somehow?

Confidence in this Review

2-Confident (read it all; understood it all reasonably well)


Reviewer 5

Summary

The authors present a novel algorithm, TRUVAR, that addresses Bayesian optimization (BO) and level-set estimation (LSE) with Gaussian processes in a unified fashion.

Qualitative Assessment

I am completely unfamiliar with the literature in Bayesian optimization and level set estimation. Unfortunately, since the paper was still assigned to me, I looked over the problem formulation and basic definitions etc. The paper seems to be clearly written, and is possibly of interest to the related community, but I have no way of judging the paper.

Confidence in this Review

1-Less confident (might not have understood significant parts)


Reviewer 6

Summary

This paper suggests a iterative elimination algorithm for Bayesian optimization that alternates between choosing a subset of promising data points and choosing observations to reduce the total (truncated) marginal variance in this subset. This framework allows experiments with different costs and works for level-set estimation. A theoretical analysis of simple regret is provided.

Qualitative Assessment

This paper is well-written and the proofs seem justifiable. It may be a borderline paper with some interesting proof techniques. However, the title seems misleading and the contribution section should be revised: instead of contrasting this paper with an arbitrary sample of previous work, what did you really do? Here is my understanding of potential contributions and my comments: 1. An algorithm based on elimination of uninteresting experiments. However, I doubt if this is novel enough by itself, because I am expecting to see more discussions on related work, e.g. [Auer 2002 using (Alg 3); Valko et al., 2014 spectral (Alg 2)]. These papers solve linear bandit systems, which are not fundamentally different from Gaussian processes. I would appreciate more discussions in the paper and in the author rebuttal to make better judgement here. 2. Being able to compare experiments that have different costs. This could be novel, as there are no obvious alternatives except explicitly writing them out in step 3 in Alg 1. Reference [20] could achieve the same but it may potentially require more computations. However, I am curious whether the simple alternative that uses GP-UCB but discounts the variance by the experiment cost is reasonable enough. 3. An analysis of the cost/sample complexity for simple regret. This analysis seems simplified because one does not need to worry about the instantaneous reward. In terms of the general result (Theorem 3.1), the cost depends on C_i, the number of experiments necessary to yield a better total posterior marginal variance on M_t, yet I am not sure how C_i actually relates to M_t. It seems to me that in Corollary 3.1, the authors fixed M_t to be the entire domain, ignoring the iterative elimination effects and compared it with the final simple regret of GP-UCB, yet the baseline GP-UCB actually considered the equivalence of elimination effects because it was originally aimed for the cumulative reward. Here is my complaint about the title. To me, level-set estimation is directly based on GP-UCB, which is already a Bayesian optimization method. While balancing exploration/exploitation, these baselines are fundamentally exploitation-major algorithms, yet this paper under review is exploration-prioritized. These method are suitable for different applications; maybe varying costs can be more easily implemented with the paper under review, but the authors need to discuss that. Also, set aside theoretical analysis, methods like expected improvements and probability of improvements can also incorporate non-uniform costs. I only read a part of the proofs but the conclusions seem very plausible. The introduction on multi-armed bandits need more work. I am not sure if heteroscedastic noise is directly related. Shrinking the variance of other points is the whole point of exploration. I expect the authors to clarify their contributions/novelty before I can make better judgement.

Confidence in this Review

2-Confident (read it all; understood it all reasonably well)